# *Akkermansia muciniphila* Is Beneficial to a Mouse Model of Parkinson’s Disease, via Alleviated Neuroinflammation and Promoted Neurogenesis, with Involvement of SCFAs

**DOI:** 10.3390/brainsci14030238

**Published:** 2024-02-29

**Authors:** Chen-Meng Qiao, Wen-Yan Huang, Yu Zhou, Wei Quan, Gu-Yu Niu, Ting Li, Mei-Xuan Zhang, Jian Wu, Li-Ping Zhao, Wei-Jiang Zhao, Chun Cui, Yan-Qin Shen

**Affiliations:** Department of Neurodegeneration and Injury, Wuxi School of Medicine, Jiangnan University, No. 1800, Lihu Avenue, Binhu District, Wuxi 214122, China

**Keywords:** *Akkermansia muciniphila*, Parkinson’s disease, inflammation, neurogenesis, SCFAs

## Abstract

Increasing evidence suggests that the gut microbiota may represent potential strategies for Parkinson’s disease (PD) treatment. Our previous research revealed a decreased abundance of *Akkermansia muciniphila* (Akk) in PD mice; however, whether Akk is beneficial to PD is unknown. To answer this question, the mice received MPTP intraperitoneally to construct a subacute model of PD and were then supplemented with Akk orally for 21 consecutive days. Motor function, dopaminergic neurons, neuroinflammation, and neurogenesis were examined. In addition, intestinal inflammation, and serum and fecal short-chain fatty acids (SCFAs) analyses, were assessed. We found that Akk treatment effectively inhibited the reduction of dopaminergic neurons in the substantia nigra pars compacta (SNpc) and partially improved the motor function in PD mice. Additionally, Akk markedly alleviated neuroinflammation in the striatum and hippocampus and promoted hippocampal neurogenesis. It also decreased the level of colon inflammation. Furthermore, these aforementioned changes are mainly accompanied by alterations in serum and fecal isovaleric acid levels, and lower intestinal permeability. Our research strongly suggests that Akk is a potential neuroprotective agent for PD therapy.

## 1. Introduction

Parkinson’s disease (PD) is the second most common neurodegenerative disorder causing significant disability in elderly individuals. It is characterized by the gradual loss as well as dysfunction of dopaminergic neurons in the substantia nigra pars compacta (SNpc), causing a decline in levels of tyrosine hydroxylase (TH), which is involved in the synthesis of levodopa in the substantia nigra-striatum system [1,2]. Genetic and environmental factors may trigger the disease [3], while neuroinflammation is thought to be a crucial element in exacerbating damage to dopaminergic neurons [4]. Interestingly, dysbiosis of the gut microbiota has been reported in PD patients [5] and animal models [6], which may worsen neuroinflammation. Identification of the beneficial gut microbiota in PD is still in the early stages. In our previous research, we found a remarkable reduction of *Akkermansia muciniphila* (Akk) in the gut of the PD mice models [7]. Whether Akk is good or not for PD needs to be demonstrated.

Akk, an anaerobic, Gram-negative bacterium, has been demonstrated to be beneficial in several diseases [8], such as ulcerative colitis [9,10], type 2 diabetes [11], non-alcoholic steatohepatitis, and associated hepatocellular carcinoma [12]. It also showed the potential to slow down the progression of Alzheimer’s disease [13], multiple sclerosis [14], and amyotrophic lateral sclerosis [15]. Our previous research has shown that the gut microbiota of mice with PD was significantly altered [7], and improving the constitution of the gut microbiome can help alleviate both neuroinflammation and inflammation of the colon [6,16]. Akk has been shown to protect the intestinal mucosa and regulate the immune system and metabolites, thus playing a beneficial role in the microbiota-gut-brain axis [17]. It is important to explore the involvement of Akk in the microbiome-brain axis in PD. Recently, we found that neuroinflammation affects adult neurogenesis [18], and neurogenesis is thought to be closely linked to changes in the brain-derived neurotrophic factor (BDNF)/tyrosine kinase receptor B (TrkB) signaling pathway [19]. In the present study, we also investigated the role of Akk in neurogenesis. As for the mechanisms of Akk, it has been reported that it may exert beneficial effects on the host through its major metabolites, which are short-chain fatty acids (SCFAs) [20,21]. Another study demonstrated that decreased SCFAs in feces and increased SCFAs in plasma in PD patients correlated with specific alterations in the gut microbiota and the extent of PD clinical severity [22]. Thus, alteration of SCFAs induced by Akk treatment, and their possible mechanism of action, were explored.

In this study, we orally administered the representative Akk strain BAA-835 to MPTP-induced subacute PD mice. Notably, Akk partially ameliorated motor dysfunction, suppressed the decrease in SNpc dopaminergic neurons, and reversed striatal TH expression. It also alleviated neuroinflammation of the striatum and hippocampus and promoted hippocampal neurogenesis, possibly through the BDNF/TrkB pathway. We tested the serum and fecal SCFAs content of mice, and found that Akk treatment affected the SCFAs content of PD mice, mainly by reducing the serum isovaleric acid content and increasing the fecal isovaleric acid content. In addition, Akk showed potential to reduce intestinal inflammatory factors, possibly through the SCFAs-G-protein coupled receptors/phosphoinositide 3-kinase/protein kinase B (SCFAs-GPCRs/PI3K/Akt) pathway. Taken together, this study demonstrated a beneficial effect of Akk in PD. We have expanded our understanding of the pathophysiological functions of Akk and offered a new potential therapeutic approach for PD clinical treatment.

## 2. Materials and Methods

### 2.1. Animal Treatment and Reagents Preparations

#### 2.1.1. Akk Culture

The Akk (ATCC BAA-835) was obtained from the American Type Culture Collection (CIP 107961, Manassas, VA, USA). It was grown in basal mucin-based brain heart infusion (BHI) medium (HB8297-4, Hopebio, Qingdao, China) at 37 °C under strict anaerobic conditions. After centrifugation at 12,000 rpm for 10 min at 4 °C, the cultures were washed and diluted with anaerobic phosphate-buffered saline (PBS) containing 8% (*v*/*v*) glycerol to an end concentration of 2 × 10^8^ CFU/mL under strict anaerobic conditions.

#### 2.1.2. Antibiotics Preparation

The antibiotic mixture was prepared as previously described [23]. The aim of the antibiotic treatment was to produce pseudo germ-free mice to avoid the inherent resistance of the mice’s own gut microbiota to probiotic colonization [24], which contributes to the colonization of the gut by exogenous Akk supplementation [25].

#### 2.1.3. 1-methyl-4-phenyl-1,2,3,6-tetrahydropyridine (MPTP, M0896, Sigma-Aldrich, St. Louis, MO, USA) Preparation

The MPTP was prepared by dissolving in a saline solution with a concentration of 3 mg/mL. The MPTP solution was injected intraperitoneally into mice at a dosage of 10 mL/kg body weight for 5 consecutive days.

### 2.2. Mouse Study Design

Group design: Male C57BL/6J mice of SPF grade (7 weeks old, weighing 22–26 g) were obtained from Beijing Vital River Laboratory Animal Technology Co. Mice (Beijing, China) were maintained with a temperature kept at 24 ± 2 °C, humidity maintained at 55 ± 10%, and a light/dark cycle set to 12 h. Food and water were freely available. Following a 7-day acclimatization, mice were assigned to 4 groups at random, including control group (Saline + Vehicle), PD model group (MPTP + Vehicle), control group with Akk gavage (Saline + Akk), and PD model group with Akk gavage (MPTP + Akk).

(i) Saline + Vehicle group: Mice were treated with antibiotics (200 μL/mouse/d, i.g) for 7 days. After 2 days of no intervention, the mice were given 5-day sterile saline (10 mL/kg/d, i.p), followed by 21-day vehicle (8% glycerol) treatment (250 μL/mouse/d, i.g);

(ii) MPTP + Vehicle group: Mice were treated with antibiotics (200 μL/mouse/d, i.g) for 7 days. After 2 days of no intervention, the mice were given 5-day MPTP treatment (30 mg/kg/d, i.p), followed by 21-day vehicle (8% glycerol) treatment (250 μL/mouse/d, i.g);

(iii) Saline + Akk group: Mice were treated with antibiotics (200 μL/mouse/d, i.g) for 7 days. After 2 days of no intervention, the mice were given 5-day sterile saline (10 mL/kg/d, i.p), followed by 21-day Akk treatment (250 μL/mouse/d, i.g);

(vi) MPTP + Akk group: Mice were treated with antibiotics (200 μL/mouse/d, i.g) for 7 days. After 2 days of no intervention, the mice were given 5-day MPTP treatment (30 mg/kg/d, i.p), followed by 21-day Akk treatment (250 μL/mouse/d, i.g).

The Jiangnan University Animal Ethics Committee approved the study’s experimental techniques (JN. No20211115c0600120[465]).

### 2.3. Behavioral Tests

Behavior training was performed on days 38–40, and testing was carried out on day 41 of the overall procedure to assess the effect of Akk on motor dysfunctions in mice with PD (Figure 1A). Double-blind methods were conducted for the behavior tests.

Pole test: The pole test was used to assess motor retardation, balance, and physical coordination in a mouse model of MPTP-induced PD [23]. A pole measuring fifty centimeters in length and one centimeter in diameter, wrapped in medical gauze, was placed inside the home cage. The mice were positioned head down at the top of the pole, and the time it took them to descend was noted. Three repetitions were done for each mouse at intervals of 15 min, and the average duration (descent time) of each group of mice was calculated.

Traction test: The traction test was performed to assess balance and muscle strength in mice [6]. The rope had a diameter of five millimeters. The mice were scored in the following ways: 4 points were awarded when they used all of their limbs to grasp the rope, 3 points for using two forelimbs and one hindlimb, 2 points for utilizing both forelimbs, and 1 point for using just one forelimb. The test was administered three times, with a 15 min break in between. The three mean scores were calculated and statistical analysis was performed.

### 2.4. Sample Collection for Feces, Serum, Brain Tissues, and Colonic Tissue

Fecal samples were collected on the 41st day of the entire process (Figure 1A). The mice defecated fresh fecal pellets in sterile cages, then the fresh pellets of feces were promptly transferred to 1.5 mL sterile EP tubes and preserved at −80 °C to preserve sample integrity until further analysis. On day 42 of the procedure, mice were executed under deep anesthesia with isoflurane (Figure 1A). For gas chromatography-mass spectrometry (GC-MS) analysis, blood was collected from mice and centrifuged at 3000 rpm at 4 °C for 10 min after 30 min of standing. The serum was then collected and immediately stored at −80 °C. For Western blot (WB) assay, total protein was rapidly extracted from fresh striatal, hippocampal, and colonic tissues separately by mixing RIPA (P0013B, Beyotime, Shanghai, China), PEI (P1081, Beyotime, China), and PMSF (ST506, Beyotime, China) at a proportion of 100:2:1 (*v*/*v*/*v*). The mixture was centrifuged at 13,000 rpm for 5 min at 4 °C to collect the supernatant after it had completed grinding. The protein content was confirmed by using the Protein Assay Kit (BL521A, Biosharp, Hefei, China). The collected proteins were preserved at −80 °C for follow-up detection. The total RNA of colon samples was obtained using the TRIzol™ substance (15596018, Invitrogen, Carlsbad, CA, USA). The total RNA purity was assessed for each sample using the NanoDropTM 1000 Spectrophotometer (840-317400, Thermo Scientific, Thermo Scientific, MA, USA, MA, USA). The dilution of the obtained total RNA was reversely transcribed to cDNA using the PrimeScript™ RT Kit (RR036A, TaKaRa, Kusatsu, Shiga, Japan) for qRT-PCR assay. For immunofluorescence (IF) and immunohistochemistry (IHC) labeling, mice were successively perfused transcardially with PBS and 4% paraformaldehyde (PFA) solution. The brains of the mice were carefully extracted and stabilized in a 4% PFA solution for 24 h. This was followed by a dehydration process, where they were first submerged in a 20% sucrose solution at 4 °C for 24 h, and subsequently in a 30% sucrose solution at the same temperature for another 24 h. Brain tissues were embedded with O.C.T. compound (4583, Sakura, Tokyo, Japan). Cross-sections of SNpc, striatum and hippocampus were cut 10 μm thick, spaced 100 µm apart with a Cryostat (CM1950, Leica, Baden-Württemberg, Germany).

### 2.5. Immunofluorescence (IF) Staining for SNpc and Hippocampus

Brain tissue slices containing SNpc (starting from bregma at −2.92 mm to approximately −3.52 mm) or hippocampal DG region (starting from bregma at −1.58 mm to approximately −3.08 mm) were immersed in sodium citrate buffer (C1010, Solarbio, Beijing, China) at a temperature of 95 °C for antigen retrieval. After three washes with PBS, sections were treated with a blocking solution (10% goat serum) for 30 min and then left overnight incubating with primary antibodies at 4 °C. The details of the primary antibodies are as follows: rabbit anti-tyrosine hydroxylase (TH, 1:1000, AB152, Merck, Kenilworth, NJ, USA), rabbit anti-doublecortin (DCX, 1:1000, 13925, Proteintech, Wuhan, China), rabbit anti-sex determining region Y-box 2 (SOX2, 1:900, 97959, Abcam, Cambridge, UK), mouse anti-glial fibrillary acidic protein (GFAP, 1:300, sc-33673, Santa Cruz Biotechnology, Paso Robles, CA, USA) and mouse anti-ionized calcium binding adaptor molecule 1 (Iba-1, 1:300, GB12105, Servicebio, Wuhan, China). The sections were washed three times in PBS and treated for 1 h at 37 °C with a secondary antibody coupled to FITC-conjugated goat anti-rabbit IgG (1:1000, A0562, Beyotime, Shanghai, China) or Cy3-conjugated goat anti-mouse IgG (1:1000, A0521, Beyotime, Shanghai, China) or goat anti-mouse IgG conjugated with 594 (33212ES60, Yeasen, Shanghai, China). Cell nuclei of the tissue were stained with fluorescence quenching anti-fade mounting solution containing DAPI (P0126, Beyotime, Shanghai, China). For the detection of TH^+^ cells, we selected 4–5 brain sections containing SNpc (Bregma −2.92 mm to approximately −3.52 mm and spaced 100 µm apart). Specifically, four brain sections containing SNpc cross sections were respectively collected from a mouse in the Saline + Vehicle group and the Saline + Akk group, and five brain sections were collected from other mice. Fluorescence images were captured with a microscopy instrument Axio Imager Z2 (Zeiss, Thuringia, Germany), and quantified using the ZEN 2.3 blue software. ImageJ analysis Software (version 1.53k) was utilized for the quantification of TH^+^ cells in SNpc (starting from bregma at −2.92 mm to approximately −3.52 mm) of right hemisphere brain sections, and the number of Iba-1^+^ cells (starting from bregma at −2.06 mm to approximately −3.08 mm) in hippocampal dentate gyrus (DG) of right hemisphere brain sections. It was also used to calculate the number of DCX^+^ cells, SOX2^+^ cells, and GFAP^+^ cells in the hippocampal DG (starting from bregma at −1.58 mm to −2.54 mm) of left hemisphere brain sections. The final statistical analysis used the mean of the number of positive cells per mouse.

### 2.6. Immunohistochemistry (IHC) Staining for Striatum

The expression of GFAP and Iba-1 in the striatum (starting from bregma at 1.10 mm to approximately 0.50 mm) was assessed by IHC analysis. The sections from each mouse were selected for GFAP or Iba-1 staining. Sections containing striatum were subjected to antigen restore as described previously. Sections were treated with 3% H_2_O_2_ (AR1108, Boster, Wuhan, China) for 20 min at room temperature to quench endogenous peroxidase. Sections were washed three times with PBS for 5 min each, then treated with 10% goat serum (SL038, Solarbio, Beijing, China) and left to stand for thirty minutes at 37 °C then incubated with either mouse anti-GFAP (1:500, MAB360, Merck, Kenilworth, NJ, USA) or mouse anti-Iba-1 (1:50, GB12105-100, Servicebio, Wuhan, China) overnight at 4 °C. The antibody binding was enhanced using a reaction amplification solution (PV9000, OriGene, Beijing, China) at a temperature of 37 °C for 20 min. After incubating for 20 min at room temperature with biotinylated goat anti-rabbit antibody (PV9000, OriGene, Beijing, China), the sections were stained for color development for 8–10 min at room temperature using an AEC (Aminoethyl Carbazole) staining kit (ZLI-9036, OriGene, Beijing, China). Finally, the slides were sealed with Water-Soluble Sealing Reagent (AR1018, Boster, Wuhan, China). Panoramic MIDI II slide scanners (3DHISTECH Ltd., Budapest, Hungary) were used to analyze all of the sections, with positive signals indicated by a dark brown color. The number of GFAP^+^ cells or Iba-1^+^ cells in the striatum at 20 × magnification was counted using CaseViewer 2.4 software.

### 2.7. Western Blotting (WB) for Striatum, Hippocampus and Colon

Protein samples were transferred from −80 °C to 4 °C for thawing. The protein sample was reacted in 5 × SDS-PAGE Sample Loading Buffer (P0015L, Beyotime, Shanghai, China) at 100 °C for 15 min. Each sample containing 24 µg of total protein was added to 10% SDS-PAGE solution. Then it was transferred to 0.45 µm PVDF membranes (IPVH00010, Merck, Kenilworth, NJ, USA). The membranes were first immersed in 5% BSA (*w*/*v*) (A8010, Solarbio, Beijing, China) for 2 h and then immersed in solutions containing different primary antibodies for 12 h at 4 °C. The information on primary antibodies is as follows: rabbit anti-glyceraldehyde-3-phosphate dehydrogenase (GAPDH, 1:8000, 10495-1-AP, Proteintech, Wuhan, China), rabbit anti-tyrosine hydroxylase (TH, 1:1000, MAB318, Millipore, MA, USA), rabbit anti-brain-derived neurotrophic factor (BDNF, 1:2000, 108319, Abcam, Cambridge, UK), rabbit anti-tyrosine kinase receptor B (TrkB, 1:1000, 4603T, CST, MA, USA), rabbit anti-phosphatidylinositol 3-kinase (PI3K, 1:1000, 4257S, CST, MA, USA), and rabbit anti-protein kinase B (Akt, 1:1000, 4691S, CST, MA, USA). The membranes were incubated with HRP-conjugated goat anti-rabbit IgG (1:8000, A0277, Beyotime, Shanghai, China) for two hours at room temperature. Membranes were finally imaged using a chemiluminescence imager (Tanon-5200Multi, Shanghai, China) after being reacted with the Super ECL Detection Reagent (36208ES60, Yeasen, Shanghai, China). The analysis was conducted using ImageJ analysis software (version 1.53k).

### 2.8. Gas Chromatography-Mass Spectrometry (GC-MS) for Serum and Feces

The content of SCFAs in the collected serum and fecal samples was analyzed by GC-MS. Serum samples were transferred from −80 °C to 4 °C for thawing, followed by vortexing for 1 min for homogenization. A quantity of 150 μL of the serum sample was acidified by mixing it with 75 μL of 50% sulfuric acid. As an internal standard, 75 μL of 0.1% 2-ethylbutyric acid (E105669, National Pharmaceutical Group Shanghai Institute, Shanghai, China) was added. After being vortexed for 1 min, the mixture was centrifuged at 12,000 rpm for 20 min at 4 °C. The supernatant was extracted and mixed with 0.2 g of anhydrous sodium sulfate to remove water. After standing for 30 min at 4 °C, the mixture was centrifuged again using the above parameters. Finally, the supernatant was analyzed for SCFAs using a Trace 1300 GC-MS system (Thermo Fisher Scientific Inc., Waltham, MA, USA) equipped with an Rtx-WAX capillary column (30 m × 0.25 mm × 0.25 μm, Restek, Bellefonte, PA, USA).

Fecal samples were processed as follows: 100 μL ultrapure water was added to 50 mg of the sample and allowed to stand for 5 min. Two magnetic beads were added for homogenization for 5 min and centrifuged at 13,000 rpm for 10 min. A quantity of 50 μL of 50% sulfuric acid was added per 100 μL of supernatant for acidification. Following mixing and 2 min of standing, the supernatant was extracted, and 225 μL of n-hexane was used to obtain organic acids by centrifugation at 14,000 rpm for 5 min. Finally, the supernatant was loaded into the GC-MS system as described above.

### 2.9. Quantitative Real-Time PCR (qRT-PCR) for Colon

qRT-PCR was performed using a Roche LightCycler 480^®^ instrument II and primer sequences validated by PrimerBank. The TB Green Premix Ex Taq ^TM^ II kit (RR820A, Takara, Kusatsu, Shiga, Japan) was used for qRT-PCR reactions. Quantification was performed using the 2^−∆∆Ct^ method, with normalization of gene expression to the GAPDH expression level. The primer sequences employed in this study are provided in Table 1.

### 2.10. Statistical Analysis

All data are reported as the mean ± standard error (SEM). Statistical comparisons were performed using one-way ANOVA followed by post hoc comparisons using LSD test for homogeneous variance and Dunnet’s T3 test for heterogeneous variance using SPSS 26.0 software. The thresholds of statistical significance were as described below: * *p* < 0.05, ** *p* < 0.01 and *** *p* < 0.001. GraphPad Prism 8.0 was utilized for conducting the statistical analyses.

## 3. Results

### 3.1. Akk Treatment Partially Improved Motor Dysfunction in MPTP-Induced PD Mice

To assess the potential effect of Akk treatment on motor dysfunction in the neurotoxin-induced mouse model, we evaluated the motor abilities of mice using the pole test and the traction test. In the pole test, the MPTP + Vehicle group showed obvious motor deficits compared to the Saline + Vehicle group (*p* < 0.01). Interestingly, mice treated with MPTP + Akk showed a significant reduction in descent time (*p* < 0.05) (Figure 1B). However, no significant alteration was observed in the scores of all groups (Figure 1C). Overall, Akk treatment could partially improve the motor dysfunction in PD mice.
Figure 1Akk partially improved motor dysfunction in MPTP-induced PD mice. (**A**) Experimental flow diagram. (**B**) Descent time in the pole test (*n* = 11–12). (**C**) Scores in the traction test (*n* = 12). Statistical comparison was conducted with one-way ANOVA followed by a Dunnet’s T3 post hoc test in (**B**) and an LSD post hoc test in (**C**). * *p* < 0.05, ** *p* < 0.01.
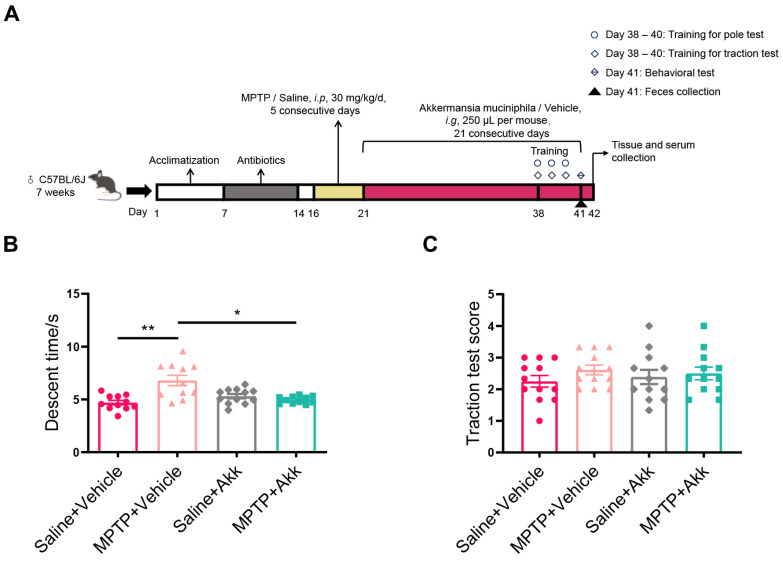



### 3.2. Akk Suppressed the Loss of Dopaminergic Neurons of SNpc and Reversed Striatal TH Expression in MPTP-Induced PD Mice

The therapeutic effect of Akk on PD mice was investigated by counting dopaminergic neurons in the SNpc and evaluating the expression level of striatal TH protein. Compared to Saline + Vehicle treated mice, the number of TH-positive cells in SNpc was significantly reduced in the MPTP + Vehicle treated group. However, MPTP + Akk treatment reversed the decrease in the number of TH-positive cells (Figure 2A,B). Similarly, TH protein expression in the MPTP + Vehicle group was significantly reduced compared to that in the Saline + Vehicle group, whereas MPTP + Akk treatment inhibited the decrease in TH protein level in the striatum (Figure 2C,D). Our data suggest that Akk suppressed the loss of dopaminergic neurons and promoted dopamine synthesis in PD mice.

### 3.3. Akk Treatment Alleviated Neuroinflammation in Striatum and Hippocampus in MPTP-Induced PD Mice

Neuroinflammation ensues as an immune response initiated by microglia and astrocytes within the central nervous system (CNS). These cells are often observed in affected brain regions during the course of PD, including the striatum and hippocampus [26]. To objectively determine the consequence of Akk on PD’s neuroinflammation, we examined the expression of GFAP^+^ cells and Iba-1^+^ cells in the aforementioned brain tissue regions. Both striatal GFAP^+^ cells and Iba-1^+^ cells were increased in the MPTP + Vehicle group, but significantly reduced in the MPTP + Akk group (Figure 3A–D). Similarly, the hippocampal DG region of MPTP + Vehicle group showed more astrocytes and microglia than those of Saline + Vehicle group, whereas the MPTP + Akk group showed decreased astrocytes and microglia numbers (Figure 4A–D). Our data suggest that Akk can alleviate neuroinflammation in PD mice.

### 3.4. Akk Treatment Promoted Hippocampal Neurogenesis in MPTP-Induced PD Mice

MPTP treatment significantly reduced the proliferation of neural stem cells (NSCs) in the subventricular zone and hippocampus [27]. The expression of SOX2^+^ cells in the hippocampal DG region may reflect the self-renewal capacity of NSCs in PD mice, and its reduction is also a significant manifestation of altered neurogenesis [28]. The amounts of DCX-stained neurons and SOX2-stained neurons in the DG of the hippocampus were detected by IF staining. As shown in Figure 5A,C, the number of DCX^+^ cells and SOX2^+^ cells in the MPTP + Vehicle treated group was markedly less than that in the Saline + Vehicle group. Compared to the MPTP + Vehicle group, mice in the MPTP + Akk group had more DCX-positive cells and SOX2-positive cells. DCX^+^ cells slightly increased in the Saline + Akk group compared to the Saline + Vehicle group, but this did not reach statistical significance. The BDNF/TrkB pathway, a significant signaling pathway in promoting the growth of new neurons, and in protecting and nourishing neurons, is involved in neurogenesis in the hippocampus [19]. WB results showed that the BDNF/TrkB signaling pathway was suppressed in the MPTP + Vehicle group compared to the Saline + Vehicle group. However, treatment with MPTP + Akk reversed this abnormal expression. Meanwhile, the Saline + Akk treatment group showed no significant difference in the expression of this pathway compared to the Saline + Vehicle treatment group (Figure 5E–H). These data suggest that Akk can promote hippocampal neurogenesis in PD mice by increasing the expression of the BDNF/TrkB neurotrophic pathway.

### 3.5. Akk Treatment Regulated Production of Serum and Fecal SCFAs, Partially Reduced Intestinal Permeability

SCFAs are the major metabolites produced by bacterial fermentation of dietary fiber in the gastrointestinal tract, and are considered to play a crucial role in microbiota-gut-brain crosstalk. As shown in Figure 6A–G, there was noteworthy alteration of isovaleric acid in both serum and fecal samples, and of butyric acid and isobutyric acid in serum samples, while there was no significant difference observed in the major SCFAs, acetic acid, propionic acid, and valeric acid in both serum and feces of mice from four groups. Compared to the Saline + Vehicle group, the MPTP + Vehicle treatment increased serum isovaleric acid levels, whereas the MPTP + Akk treatment inhibited the abnormal increase in isovaleric acid levels. In fecal samples, although the MPTP + Vehicle treatment had no significant effect on isovaleric acid levels compared to the Saline + Vehicle group, the MPTP + Akk treatment significantly increased isovaleric acid levels in the MPTP + Vehicle group mice. Similarly, the MPTP + Akk group significantly increased fecal isovaleric acid levels compared to the Saline + Akk group (Figure 6G). Butyric acid markedly decreased in the MPTP + Akk group compared to the MPTP + Vehicle group, although the MPTP + Vehicle group showed no obvious change compared to the Saline + Vehicle group (Figure 6D). In addition, the Saline + Akk treatment significantly reduced isobutyric acid levels in the Saline + Vehicle group, although there was no significant difference in isobutyric acid levels compared to other groups (Figure 6E). Our data show that Akk can inhibit the abnormal elevation of serum isovaleric acid levels in PD mice, while simultaneously increasing fecal isovaleric acid levels.

The permeability of the gut-blood barrier (GBB) was assessed by calculating the ratio of the concentration of each SCFA in the serum of each mouse to the concentration of the SCFA in feces (C_S_/C_F_) [29,30]. As shown in Figure 6H, there was no significant change in the C_S_/C_F_ of the major SCFAs or the six SCFAs mentioned above between the MPTP + Vehicle group and the Saline + Vehicle group. However, the MPTP + Akk treatment significantly decreased the C_S_/C_F_ of isobutyric acid and isovaleric acid compared to the MPTP + Vehicle group. Taken together, these results suggest that Akk treatment led to a reduction in intestinal permeability in PD mice to some extent, which is consistent with previous research suggesting that Akk improves intestinal barrier function [9,11,21].

### 3.6. Effect of Akk Treatment on Intestinal Inflammation in MPTP-Induced PD Mice

Recent studies have provided evidence that PD results from an alteration in the gut microbiota, characterized by changes in gut permeability due to an increase in pathogenic bacteria [31]. Compared to the Saline + Vehicle group, the MPTP + Vehicle group showed increased expression of the proinflammatory cytokines IL-12a in mice, although it showed no significant difference in expression. In contrast, the MPTP + Akk treatment group showed a significant decrease in the expression of IL-12a and TLR4 compared to the MPTP + Vehicle group (Figure 7A,B). These results indicated that Akk could reduce the expression levels of intestinal inflammatory related factors. In previous studies, isovaleric acid has been shown to activate GPR41/43 [32], and that both the GPR41/43 signaling pathway [33] and the PI3K/Akt signaling pathway (downstream of GPR41) [34] are involved in the anti-inflammatory response in the intestine. Since our experimental results showed that Akk increased the level of isovaleric acid in the gut, we further examined the expression of GPR41/43 and the PI3K/Akt pathway in the colon. We confirmed GPR41 and GPR43 levels were remarkably higher in the MPTP + Akk group than the MPTP + Vehicle group, although the MPTP + Vehicle group showed no obvious change in GPR41 and GPR43 compared to the Saline + Vehicle group (Figure 7C,D). Our data indicated that isovaleric acid activated the corresponding GPR41/43 receptors in the gut. Furthermore, the PI3K/Akt pathway showed a significant decrease in the MPTP + Akk treatment group (Figure 7E,F), suggesting that Akk may reduce the expression of intestinal inflammatory factors by activating both GPR41 and GPR43 and suppressing the downstream PI3K/Akt pathway.

## 4. Discussion

The gut-brain axis is a two-way communication pathway connecting the gastrointestinal tract and the CNS. It involves the autonomic nervous system, enteric nervous system, neuroendocrine system, and immune system [35]. Recent studies have shown a strong correlation between gut inflammation and gut microbiota dysbiosis, which are involved in the gut-brain axis and the development of PD [6,7,16,23,36]. As an adjunctive treatment for PD, probiotic supplementation may improve PD conditions by altering the microbiota associated with PD, including gut inflammation and CNS function [37].

Akk is a representative microbial community in the human gut with a unique ability to degrade mucins. It uses mucins as its sole carbon and nitrogen source and exhibits strong adhesion and colonization in the gut [38]. It has been reported to have significant and strong associations with metabolic disorders, specific gastrointestinal diseases, and some neurodegenerative diseases [39]. For example, Akk increased the ability to reduce fat accumulation and insulin resistance in mice [40]. It also enhances intestinal barrier function and improves colon length shortening and histopathological scores in dextran sulfate sodium-induced colitis [10]. In addition, Akk has been shown to attenuate cognitive impairment and brain amyloid-beta deposition in an APP/PS1 mice model of Alzheimer’s disease [13]. Our previous study showed that the reduction of dopaminergic neurons in mice after MPTP modeling was accompanied by a decrease of Akk abundance in the intestine [7], which is consistent with the vast majority of studies to date [41,42,43]. However, some studies have observed an increase in Akk in the gut of PD patients [44,45,46]. The possible reason for this phenomenon is that different strains of the same bacteria exhibit different effects [47]. Recently, Fang et al. found that an increase in the abundance of Akk after administration of an engineered strain, MG1363-pMG36e-GLP-1, ameliorated the symptoms and pathology of PD [48], which is similar to our previous results with vancomycin-pretreated PD mice [36] and PD mice that received microbiota transplants from aged mice [7]. The above work suggests that Akk is strongly associated with the pathogenesis of PD, but no research has demonstrated the therapeutic effects of Akk treatment alone on PD mice and elucidated its mechanism on the gut-brain axis.

Since the mouse’s own microbiota can develop inherent resistance to the colonization of probiotics [24], while antibiotic pretreatment facilitates the colonization and enrichment of Akk supplements in the gut [25], we administered antibiotic pretreatment to the mice to manifest the effect of Akk alone. We found that oral Akk intake was able to prevent the loss of dopaminergic neurons in PD mice, accompanied by restoration of striatal TH expression, suggesting a beneficial role of Akk in PD mice. Neuroinflammation is an important pathogenesis of PD, and we show that Akk has anti-inflammatory effects in both the striatum and hippocampus; specifically, in these brain regions, Akk treatment significantly reduced the expression of astroglia-specific marker (GFAP) and microglia-specific marker (Iba-1). Hippocampal dentate granule cells in PD patients show abnormal morphogenesis and changes in the expression of differentiation markers [49], and neuroinflammation has been shown to alter adult neurogenesis [18,50,51]. Here, we show that Akk attenuates neuroinflammation in PD mice and increases the expression of DCX and SOX2 markers in the mouse hippocampus DG region, so we reasonably speculate that the improvement of neuroinflammation may be the reason why Akk promotes neurogenesis. Inhibition of hippocampal neurogenesis in PD model mice is associated with downregulation of the BDNF/TrkB pathway [19], and our data suggest that Akk promotes the restoration of hippocampal neural stem cell marker expression, which could be associated with the upregulation of the BDNF/TrkB pathway. We firstly report that Akk attenuates neuroinflammation and promotes hippocampal neurogenesis in PD mice. 

Most studies have shown that SCFAs have various local effects that can improve gut health and directly or indirectly affect the brain [35,52,53,54]. We found that Akk improves intestinal barrier function, possibly through SCFAs, thereby maintaining the stability of the gut microbiota. SCFAs can also be used as interventional agents to target human microbe-gut-brain interactions [55]. For example, SCFAs have been shown to rescue the decline of hippocampal neurogenesis in mice including the reduction of DCX expression [56]. In addition, amelioration of neuroinflammation in mice was associated with increased fecal isovaleric acid levels and improved gut barrier function [52]. Interestingly, in addition to the direct suppression of neuroinflammation and promotion of neurogenesis, we also provided the first evidence that Akk modulates serum and fecal isovaleric acid production in PD model mice. The gut microbiota can influence the nervous system by releasing metabolites like SCFAs, which can alter the permeability of the blood-brain barrier [57,58] and easily cross it [59]. Akk reduced serum isovaleric acid levels in PD mice, which may be directly involved in the process of Akk’s positive effect on the brain. Based on these findings, we suggest that Akk may exert neuroprotective effects in PD mice through isovaleric acid mediation. Based on the methodology of Jaworska et al. [30], we obtained mouse GBB permeability by calculating the ratio of each SCFA concentration in serum and feces (C_S_/C_F_), and the decrease of this permeability can indirectly indicate the enhancement of intestinal barrier function. Our data suggest that Akk may play a role in improving the integrity and permeability of intestinal microecology. In addition, Akk suppressed the expression of colonic inflammatory factors in PD mice; we further explored its potential mechanism, and found that Akk increased the level of isovaleric acid in the feces of PD mice, and also that isovaleric acid may further up-regulate the expression of SCFAs-related receptors GPR41 and GPR43 in the colon. Moreover, the PI3K/Akt pathway in the colon of PD mice was further down-regulated, suggesting that Akk may provide potential intervention strategies to reduce colonic inflammation in PD mice through the SCFAs-GPCRs/PI3K/Akt pathway. Ishii T et al. suggest that luminal gut *Bifidobacterium breve* strain A1 may transmit to the SNpc via gut-innervating vagal sensory neurons, increasing dopaminergic neurons in the SNpc and inhibiting abnormal changes in hippocampal synaptic plasticity via a still-unidentified neural mechanism [60]. Based on these results, we speculate that Akk might affect the changes of neurons in SNpc, hippocampus inflammation, and stem cell-related markers in PD mice, by regulating the production of isovaleric acid in the gut and further through the unknown signal of the microbiome-gut-brain axis.

## 5. Conclusions

To sum up, we have demonstrated the potential neuroprotective effects of Akk in a MPTP-induced subacute PD mouse model. These effects include improvements in motor deficits, brain pathology such as neuroinflammation and neurogenesis, and colonic inflammation in PD mice. These findings were accompanied by changes in fecal and serum isovaleric acid levels. Furthermore, it is reasonable to hypothesize that isovaleric acid may have potential neuroprotective effects on the pathology and symptoms of PD. It will be crucial to explore the effects of isovaleric acid on PD models in vitro and in vivo in the future.

## Figures and Tables

**Figure 2 brainsci-14-00238-f002:**
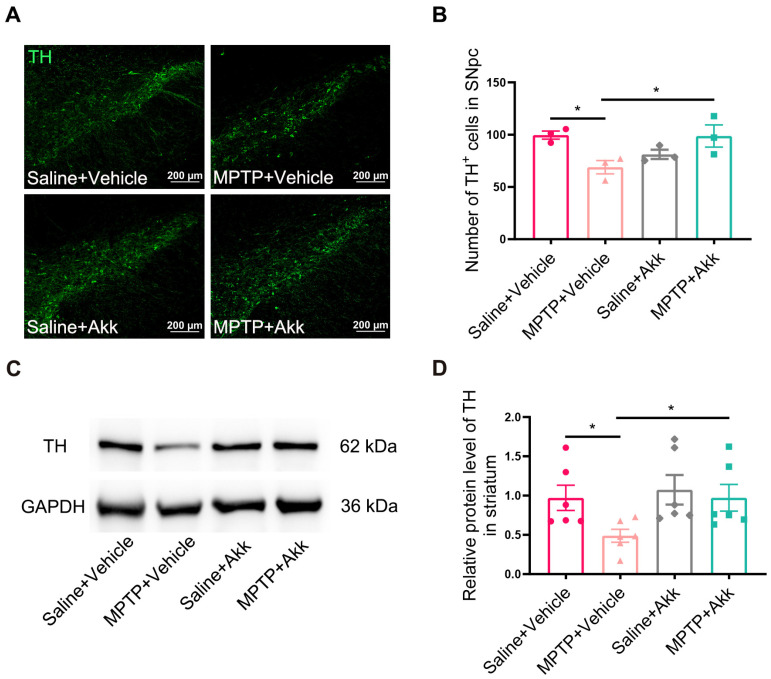
Akk inhibited the loss of dopaminergic neurons in the SNpc of PD mice. (**A**) The representative IF images of dopaminergic neurons in the SNpc, scale bar is 200 μm. (**B**) The number of dopaminergic neurons in the SNpc (*n* = 3). (**C**) The representative WB images of TH and GAPDH expression in striatum. (**D**) The intensity of the bands was measured using ImageJ analysis software (version 1.53k) and normalized with GAPDH (*n* = 6). Statistical comparison was conducted with one-way ANOVA followed by post hoc comparisons of LSD. * *p* < 0.05.

**Figure 3 brainsci-14-00238-f003:**
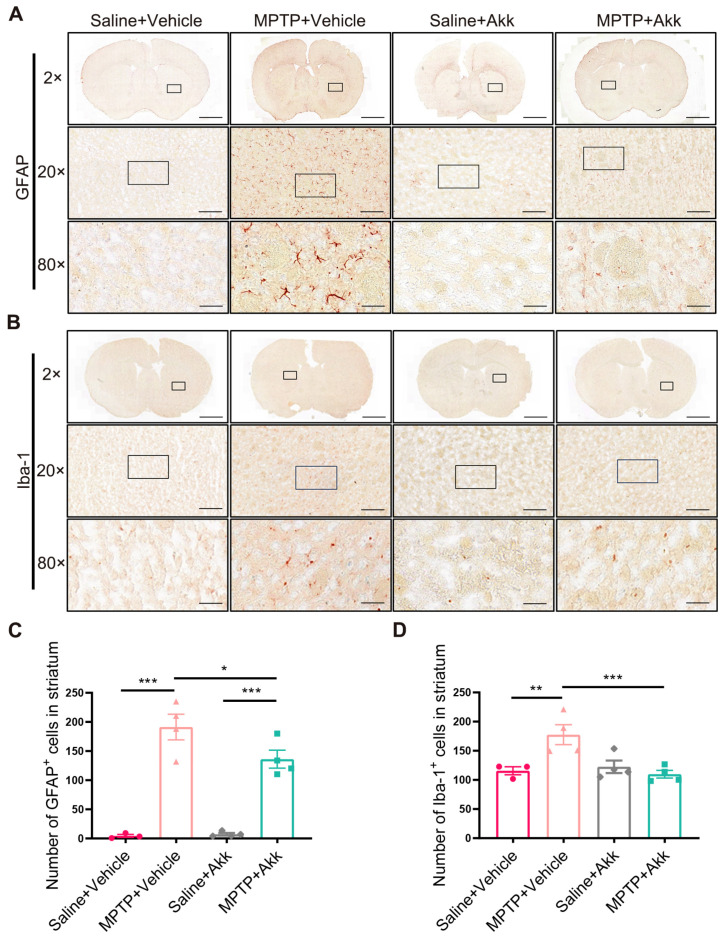
Akk alleviated neuroinflammation in the striatum of PD mice. (**A**,**B**) The representative images of IHC for GFAP^+^ cells and Iba-1^+^ cells in the striatum. The high magnification of images in the black box is shown below. Scale bars at 2×, 20×, and 80× are respectively 2 mm, 200 μm, and 50 μm. (**C**,**D**) Number of GFAP^+^ cells and Iba-1^+^ cells in the striatum under a 20× field of view using CaseViewer 2.4 software (*n* = 3–4). Statistical comparison was conducted with one-way ANOVA followed by post hoc comparisons of LSD. * *p* < 0.05, ** *p* < 0.01, *** *p* < 0.001.

**Figure 4 brainsci-14-00238-f004:**
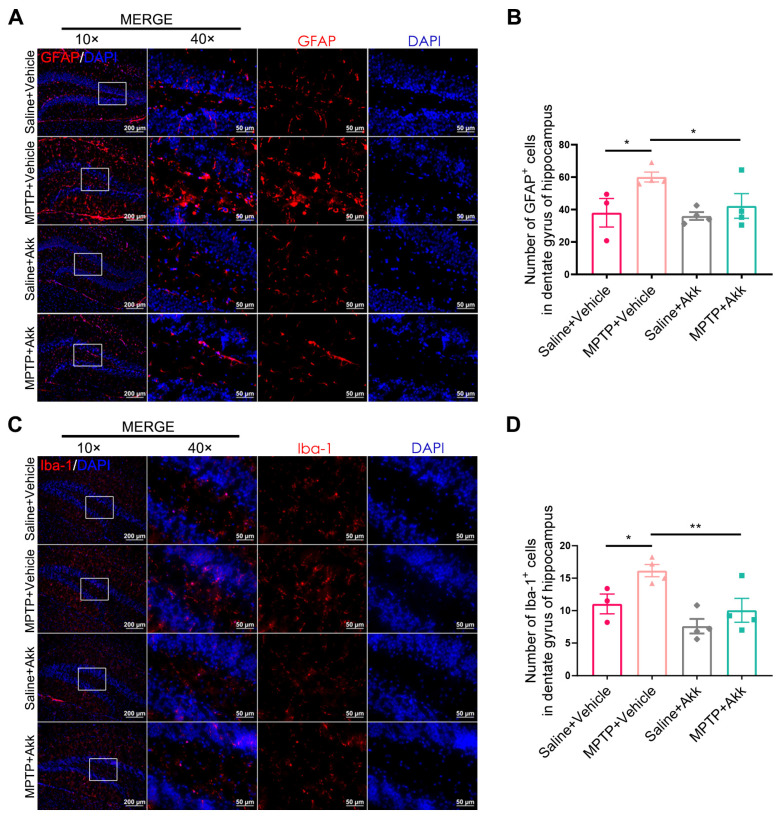
Akk alleviated neuroinflammation in the hippocampus of PD mice. (**A**) Representative IF images of GFAP expression in the hippocampus. The scale bar is 200 μm. The high magnification of images is shown in the white box on the right. The scale bar is 50 μm. (**B**) The number of GFAP^+^ cells in the hippocampal DG region at 10 × magnification (*n* = 3–4). (**C**) Representative IF images of Iba-1 expression in the hippocampus. The scale bar is 200 μm. The high magnification of images is shown in the white box on the right. The scale bar is 50 μm. (**D**) The number of Iba-1^+^ cells in the hippocampal DG region at 10 × magnification (*n* = 3–4). Statistical comparison was conducted with one-way ANOVA followed by post hoc comparisons of LSD. * *p* < 0.05, ** *p* < 0.01.

**Figure 5 brainsci-14-00238-f005:**
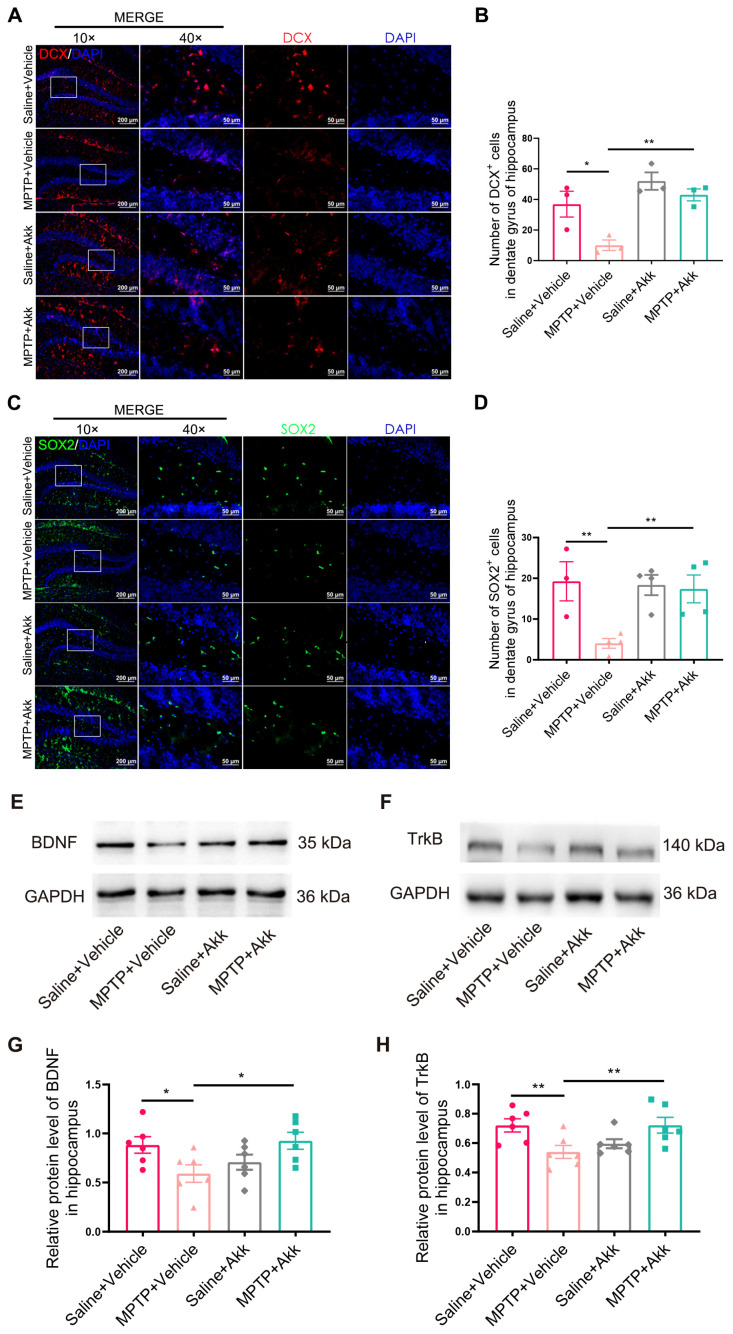
Akk promoted neurogenesis in the hippocampus of PD mice, possibly through activating the BDNF/TrkB signaling pathway. (**A**) Representative IF images of DCX-positive cells (immature neuron marker). Scale bar = 200 μm. The high magnification of images is shown in the white box on the right. The scale bar is 50 μm. (**B**) Quantitative analysis of DCX-positive cells in the hippocampal DG region at 10× magnification (*n* = 3). (**C**) Representative IF images of SOX2-positive cells (neural stem cell marker). Scale bar = 200 μm. The high magnification of images is shown in the white box area on the right. The scale bar is 50 μm. (**D**) Quantitative analysis of SOX2-positive cells in the hippocampal DG region at 10× magnification (*n* = 3–4). (**E**,**F**) The representative WB images of BDNF and TrkB expression in the hippocampus. (**G**,**H**) WB analysis of BDNF/GAPDH ratio and TrkB/GAPDH ratio in hippocampus (*n* = 6). Statistical comparison was conducted with one-way ANOVA followed by post hoc comparisons of LSD. * *p* < 0.05, ** *p* < 0.01.

**Figure 6 brainsci-14-00238-f006:**
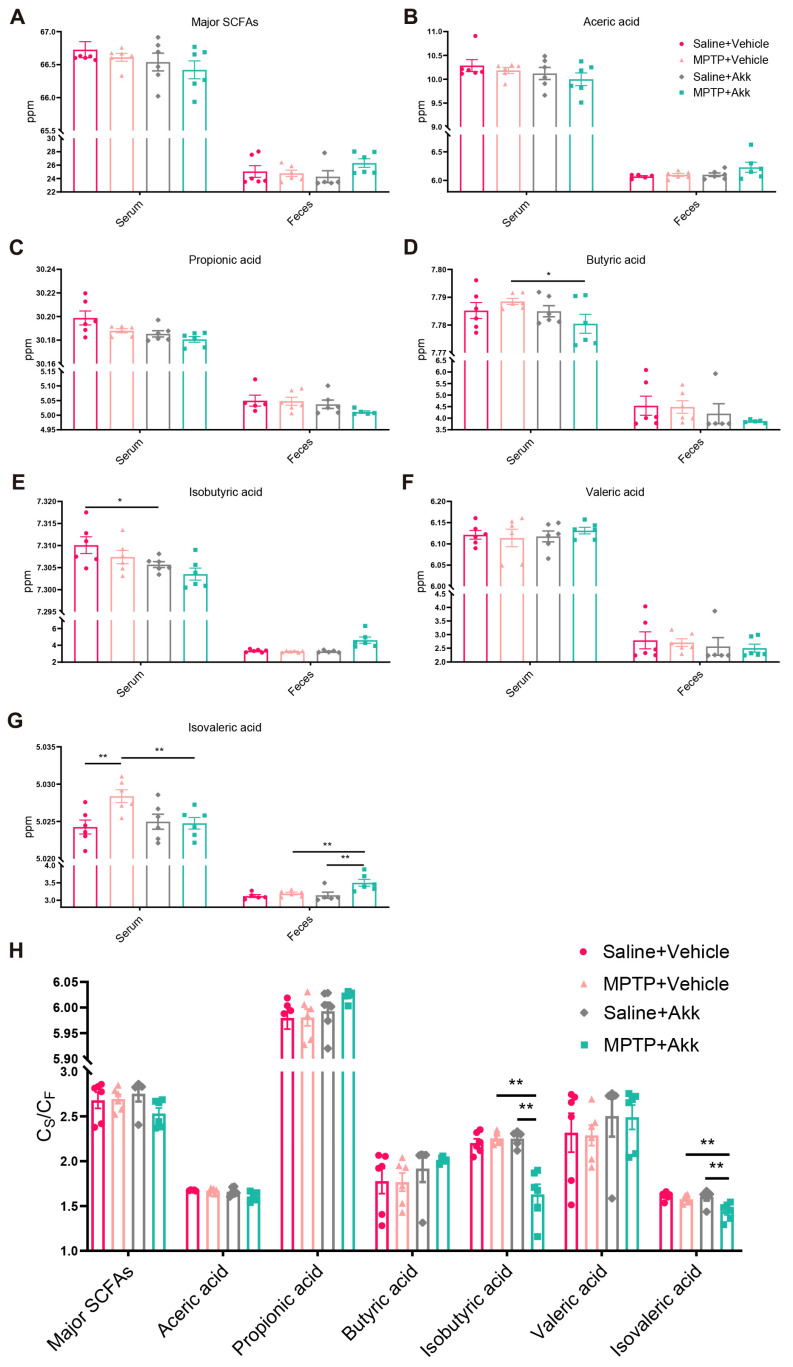
The concentrations of SCFAs in mouse serum and feces, as well as their ratios (C_S_/C_F_). (**A**–**G**) The concentrations of the major SCFAs in both serum and feces, including acetic acid, butyric acid, valeric acid, propionic acid, isobutyric acid, and isovaleric acid, as well as the total concentration of the above SCFAs. (**H**) The ratio of serum to fecal SCFAs (*n* = 5–6). Statistical comparison was conducted with one-way ANOVA followed by post hoc comparisons using LSD test for homogeneous variance or Dunnet’s T3 test for heterogeneous variance. * *p* < 0.05, ** *p* < 0.01.

**Figure 7 brainsci-14-00238-f007:**
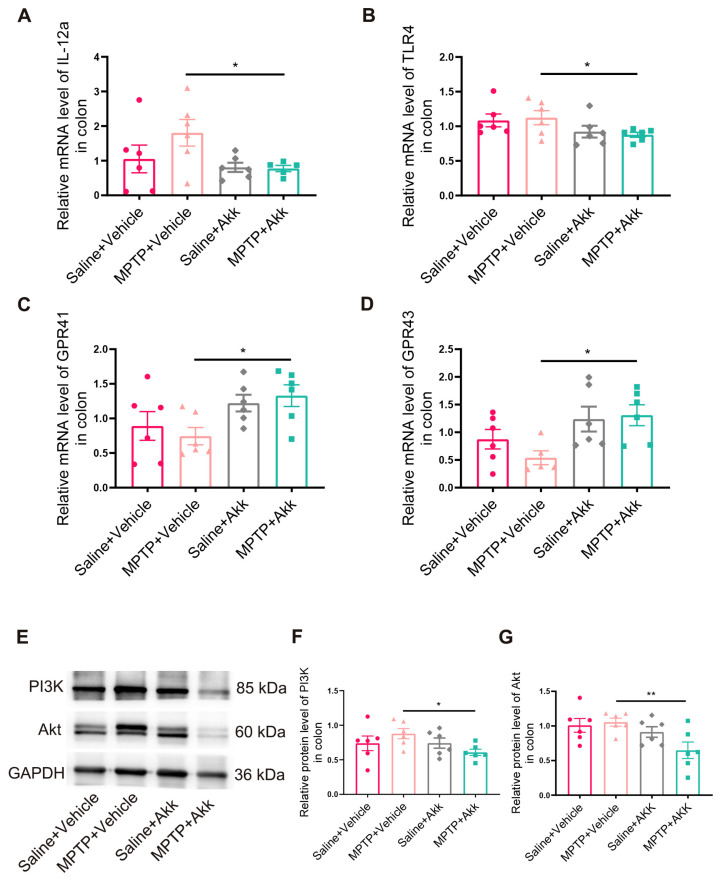
Akk activated GPR41 and GPR43, inhibited the PI3K/Akt inflammatory pathway, and relieved colonic inflammation in PD mice. (**A**,**B**) Relative mRNA levels of the inflammatory related cytokines IL-12a and TLR4 in the colon (*n* = 5-6). (**C**,**D**) Relative mRNA levels of GPR41 and GPR43 (*n* = 5–6). (**E**) The representative WB images of PI3K/Akt expression in the colon. (**F**,**G**) Quantitative analysis of PI3K/Akt protein expression in the colon (*n* = 6). Statistical comparison was conducted with one-way ANOVA followed by post hoc comparisons of LSD. * *p* < 0.05, ** *p* < 0.01.

**Table 1 brainsci-14-00238-t001:** The primer sequences utilized in qRT-PCR.

Genes	Forward and Reverse Sequences
*Gapdh*	Forward: 5′-AGGTCGGTGTGAACGGATTTG-3′ Reverse: 5′-TGTAGACCATGTAGTTGAGGTCA-3′
*Gpr41*	Forward: 5′-CTTCTTTCTTGGCAATTACTGGC-3′ Reverse: 5′-CCGAAATGGTCAGGTTTAGCAA-3′
*Gpr43*	Forward: 5′-CTTGATCCTCACGGCCTACAT-3′ Reverse: 5′-CCAGGGTCAGATTAAGCAGGAG-3′
*Il-12a*	Forward: 5′-CTGTGCCTTGGTAGCATCTATG-3′ Reverse: 5′-GCAGAGTCTCGCCATTATGATTC-3′
*Tlr4*	Forward: 5′-ATGGCATGGCTTACACCACC-3′ Reverse: 5′-GAGGCCAATTTTGTCTCCACA-3′

## Data Availability

The data that support the findings of this study are available from the corresponding author. The data are not publicly available due to ethical and privacy constraints.

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
