# Peer review of "Akkermansia muciniphila Is Beneficial to a Mouse Model of Parkinson’s Disease, via Alleviated Neuroinflammation and Promoted Neurogenesis, with Involvement of SCFAs"

_brainsci, 2024, doi:10.3390/brainsci14030238_

Round 1

Reviewer 1 Report

Comments and Suggestions for Authors

The article submitted by Chen Meng Qiao entitled Akkermansia muciniphila is Beneficial to Mouse Model of Parkinson’s Disease, via Alleviated Neuroinflammation and Promoted Neurogenesis, with Involvement of SCFAs" is a nice piece of work on MPTP induced mice model of PD. The authors have shown beneficial effects of AKK which alleviated neuroinflammation and decreased motor symptoms and increased neurogenesis. Overall, AKK showed its neuroprotective potential.

Comments on the Quality of English Language

Minor editing of English language required

Reviewer 2 Report

Comments and Suggestions for Authors

The manuscript presents the effects of Akkermansia muciniphila on parkinsonism and neurodegeneration induced by MPTP administration in mice.  The manuscripts' content is interesting, but several points must be addressed to adequate presentation. 

a) Introduction should include all data approached in your work. Some details justifiying a methodology are in methods instead of at this section. 

b) Check carefully the methodology declarations. For example: you declared a dose of 30mg/Kg/day in a solution of 3mg/mL...it means you should administered 10mL in one day. Is it right?

c) Be clear about the selection of motor-behavioral test used in your evaluation. Some key data, as those related to coordination are poorly explored in grasp or pole tests. Additional test were used? They should be declared; or justify the limitation to these tests? Tremor was present in the MPTP-control group? Tremor was observed in all the mice? It was not presented?

c) Discussion is too brief. It could be enriched with data of bacteria inducing changes in the same markers which were explored or measured in this work. Also, you could add data regarding the different pathways involved in the putative action to neuroprotection and local changes (in intestine).

d)   Additional data supporting relationship among data from SNc, Hippocampus and intestine should be described. Why not additional elements of the direct and indirect pathways from SNc to Thalamus were explored? Why the motor cortex or cerebellum were no described as sites where essential changes could be found?

e) The entire manuscript should be revised to avoid grammar and technical mistakes limiting the presentation. In this sense, limitation of events and restoration are different processes. How AkM could 'restore' protein expression or number of neurons? Neuroprotective agent, neuromodulator, agonist of X-event, and inducer of neurogenesis should be carefully used.

Comments on the Quality of English Language

The manuscript must be revised to avoid thecnical and grammar mistakes.

Reviewer 3 Report

Comments and Suggestions for Authors

The study investigates the potential therapeutic role of Akk in Parkinson’s disease treatment. Using a subacute PD mouse model induced by MPTP, the researchers orally administered Akk for 21 days and evaluated various parameters including motor function, dopaminergic neurons, neuroinflammation, neurogenesis, intestinal inflammation, and SCFAs. Results indicate that Akk treatment effectively restored dopaminergic neuron numbers in the SNpc, partially improved motor function, alleviated neuroinflammation in the striatum and hippocampus, and promoted hippocampal neurogenesis. Furthermore, Akk decreased colon inflammation, accompanied by alterations in serum and fecal isovaleric acid levels and reduced intestinal permeability. The findings suggest that Akk may serve as a potential neuroprotective agent for PD therapy, highlighting its therapeutic potential in mitigating PD-related pathology.

1)    The overall writing has some formatting issues, like wording and spacing. I suggest the authors check the grammar and avoid any typos. More importantly, the writing needs improvement.

2)    The experimental method part is lack of details. More detailed descriptions are needed to explain the detailed experimental procedures.

3)    I would recommend the authors to polish figures with high resolution and consistent font size.

4)    I recommend the authors to include some discussions on related studies investigating drug metabolic function (PMID: 33461059; PMID: 35173534), which helps expand the scope of the study.

Comments on the Quality of English Language

 The overall writing has some formatting issues, like wording and spacing. I suggest the authors check the grammar and avoid any typos. More importantly, the writing needs improvement.

Round 2

Reviewer 2 Report

Comments and Suggestions for Authors

Authors have addressed all my comments and suggestions. The article could be accepted. 

Comments on the Quality of English Language

Minor grammar mistakes remain.

Reviewer 3 Report

Comments and Suggestions for Authors

I have no more concerns on this manuscript.